# Relationship between Fasting and Postprandial Glucose Levels and the Gut Microbiota

**DOI:** 10.3390/metabo12070669

**Published:** 2022-07-20

**Authors:** Yui Mineshita, Hiroyuki Sasaki, Hyeon-ki Kim, Shigenobu Shibata

**Affiliations:** Laboratory of Physiology and Pharmacology, School of Advanced Science and Engineering, Waseda University, Tokyo 162-0056, Japan; m-yk_1426@fuji.waseda.jp (Y.M.); hiroyuki-sasaki@asagi.waseda.jp (H.S.); gusrlgusrl82@gmail.com (H.-k.K.)

**Keywords:** postprandial hyperglycemia, gut microbiota, postprandial blood glucose

## Abstract

Postprandial hyperglycemia increases the risk of mortality among patients with type 2 diabetes and cardiovascular diseases. Additionally, the gut microbiota and type 2 diabetes and cardio-vascular disease are known to be correlated. Currently, fasting blood glucose is the primary in-dex for the clinical diagnosis of diabetes; however, postprandial blood glucose is associated with the risk of developing type 2 diabetes and cardiovascular disease and mortality. Therefore, the dynamic change in blood glucose levels under free-living conditions is considered an important and better marker than fasting glucose levels to study the relationship between glucose levels and microbiota. Here, we investigated the relationship between fasting and postprandial glucose levels and microbiota under free-living conditions for one week in older adults. In addition, in order to clarify the relationship between blood glucose level and intestinal bacteria, postprandial 4-h AUC was calculated and the correlation with gut bacteria was investigated. As a result of the present study, we observed many of the most significant correlations between the gut bacteria and the peak glucose levels after dinner and the 4-h AUC after dinner. Together, these findings suggest that the individual pattern of microbiota may help to predict post-dinner hyperglycemia and the risk of abnormal glucose metabolism, such as diabetes.

## 1. Introduction

The microbes in our bodies collectively make up approximately 100 trillion cells, which is 10 times the number of human cells, and they have 100 times more endemic genes than the human genome [1]. Most of the microbes reside in the gut and are collectively called “gut microbiota” [2]. The gut microbiota has a profound influence on human physiology and nutrition and is crucial for human life [3,4]. Previous studies, using the 16S ribosomal ribonucleic acid (rRNA) gene sequence, showed that two bacterial phyla, Bacteroidetes and Firmicutes, constitute over 90% of the known phylogenetic categories and dominate the distal gut microbiota [5,6]. Notably, the gut microbiota is extremely diverse, even among healthy people [6,7,8,9]. In recent years, several studies have examined gut microbiota using next-generation sequencing and shown that changes in the gut microbiota may be associated with metabolic diseases, including type 2 diabetes, obesity, and cardiovascular diseases [10,11,12,13]. However, reports on the association between diseases and certain taxa are inconsistent. In addition, postprandial hyperglycemia increases the risk of metabolic diseases and mortality, even in people with normal fasting blood glucose levels [14,15]. Currently, fasting blood glucose is used as a diagnostic marker for diabetes; however, given that postprandial blood glucose is associated with the risk of developing type 2 diabetes and cardiovascular disease and mortality, blood glucose levels under free-living conditions are considered more important. The association between different types of glucose levels (fasting and/or postprandial blood glucose levels) and the gut microbiota remains largely unknown and assessing this relationship could aid in predicting the onset of type 2 diabetes and cardiovascular disease.

In this study, we examined the correlation between gut microbiota and postprandial blood glucose in each meal, which is more associated with the risk of developing type 2 diabetes, cardiovascular disease, and mortality in comparison with fasting blood glucose under free-living conditions. By examining these correlations, we aimed to predict not only postprandial glucose levels but the risk of abnormal glucose metabolism, such as developing type 2 diabetes.

## 2. Materials and Methods

### 2.1. Study Participants

This study, conducted between July and September 2018, included healthy older adults (*n* = 30; 14 men and 16 women), aged 65 and over (74.2 ± 5.29), from Tokyo (Japan); the inclusion criteria were as follows: (1) not receiving any antioxidant, anti-obesity, or anti-diabetic supplements; (2) no diagnosis of diabetes, dyslipidemia, or sleep apnea syndrome by a physician; (3) no hypertension (systolic blood pressure > 140 mmHg, diastolic blood pressure > 90 mmHg); and (4) the absence of the use of glucose/insulin-lowering or related medications. All participants filled a questionnaire on dietary habits, lifestyle habits, and health and medication status prior to the study. Eight participants were excluded from the study owing to the submersion of their feces during experiments.

### 2.2. Study Design

The experiments were conducted for a week, and participants were asked to maintain their normal life without changing their lifestyle habits, such as diet and exercise. During the experimental period, the physical characteristics of all participants were measured, and all subjects were asked to wear a continuous glucose monitoring system. They were also asked to collect their feces in a tube with phosphate-buffered saline containing 20% glycerol, which had been distributed in advance after the experimental period for intestinal microbiota evaluation (morning of Day 8). The collected fecal samples were transported to the laboratory at 4 °C and were then immediately frozen in liquid nitrogen and stored at −80 °C until analysis.

## 3. Measurements

### 3.1. Anthropometry

Body mass was measured to the nearest 0.1 kg using a digital balance (Inbody 230, Inbody Inc., Tokyo, Japan) and height was measured to the nearest 0.1 cm using a wall-mounted stadiometer (YS-OA, As One Corp., Osaka, Japan). Body mass index (BMI) was calculated as weight in kilograms divided by the square of height in meters, while muscle mass was measured by direct segmental multifrequency (20 kHz to 100 kHz) bioelectrical impedance (Inbody 230, Inbody Inc., Tokyo, Japan).

### 3.2. Determination of Interstitial Fluid Glucose Levels

All subjects were required to wear a continuous glucose monitoring system (FreeStyle Libre Pro; Abbott Laboratories, Chicago, IL, USA) for the continuous measurement of interstitial fluid glucose levels during the intervention. When worn, the system can continuously measure and store interstitial fluid glucose levels at 15 min intervals and is considered to be less burdensome than other glucose monitoring systems, even for elderly people. The sensor was worn on the back of the upper arm. The parameters used to evaluate glycemic variability were standard deviation (SD), coefficient of variation (CV), and peak glucose levels.

### 3.3. Fecal DNA Extraction and 16S rRNA Gene Sequencing

T16S rRNA gene sequencing was performed on the Illumina platform. The V3-V4 variable regions of the 16S rRNA gene were amplified via PCR using the following primers:

Forward Primer: 5′-TCGTCGGCAGCGTCAGATGTGTATAAGAGACAGCCTACGGGNGGCWGCAG-3′;

Reverse Primer: 5′-GTCTCGTGGGCTCGGAGATGTGTATAAGAGACAGGACTACHVGGGTATCTAATC-3′.

Amplicon PCR was performed with 2.5 µL of microbial DNA (5 ng/µL), 5 µL of each primer (1 µM), and 12.5 µL of 2× KAPA HiFi HotStart Ready Mix (Kapa Biosystems, Wilmington, MA, USA). The cycling parameters were as follows: denaturation at 95 °C for 3 min, followed by 25 cycles of denaturation, annealing, and elongation at 95 °C for 30 s, 55 °C for 30 s, and 72 °C for 30 s, respectively, and a final extension at 72 °C for 5 min. PCR amplicons were purified using AMPure XP beads (Beckman Coulter Inc., Brea, CA, USA).

To perform multiplex sequencing, adapters and barcodes were ligated to amplicons using the Nextera XT Index Kit v2 (Illumina Inc., San Diego, CA, USA). Index PCR was performed with 5 µL PCR product, 5 µL of each Nextera XT Index primers, 25 µL of 2× KAPA HiFi HotStart Ready Mix, and 10 µL of PCR-grade water under the following conditions: one cycle at 95 °C for 3 min, eight cycles of denaturation, annealing, and extension at 95 °C for 30 s, 55 °C for 30 s, and 72 °C for 30 s, respectively, followed by a final extension at 72 °C for 5 min. The quality of the purified products was evaluated using an Agilent 2100 Bioanalyzer with a DNA 1000 kit (Agilent Technologies, Santa Clara, CA, USA). Finally, the DNA library was diluted to a concentration of 4 nM and sequenced using MiSeq Reagent Kit v3 (Illumina Inc., San Diego, CA, USA) on an Illumina MiSeq 2 × 300 bp platform, according to the manufacturer’s instructions.

### 3.4. Analysis of 16S rRNA Gene Sequences

The 16S rRNA sequence reads were processed using quantitative insights into the microbial ecology (QIIME, http://qiime.org/, accessed on 20 July 2022.) pipeline version 1.9.1 [16]. The quality-filtered sequence reads were assigned to operational taxonomic units (OTUs) using closed-reference OTU picking at 97% identity with the UCLUST algorithm [17]. These reads were then compared with reference sequence collections in the Greengenes database (August 2013 version). In total, 1,296,946 reads were obtained from 44 samples, and on an average, 22,361 ± 2,721 reads were obtained per sample. The taxonomy summary at the phylum and genus levels was calculated using the QIIME software (version 1.9.1).

### 3.5. Statistical Analysis

Data were analyzed using the Predictive Analytics Software for Windows (SPSS Japan Inc., Tokyo, Japan). All parameters were tested for normal or non-normal distributions using the Kolmogorov–Smirnov test. In this study, the correlation between fasting (1 point before breakfast) and postprandial (breakfast, lunch, and supper) glucose levels and the gut bacteria was investigated. In addition, in order to clarify the relationship between blood glucose level and intestinal bacteria, postprandial (breakfast, lunch, and supper) a 4-h Area Under the Curve (AUC) was calculated and the correlation with gut bacteria was investigated. The correlation coefficient was calculated using Pearson’s or Spearman’s test for the parameters showing either a normal or non-normal distribution, respectively. Gut bacteria found in more than half of the subjects were selected and used for statistical analysis (50 types of gut bacteria from ≥11 persons were used). Levene’s test was used to compare the variance in fasting and peak (after each meal) glucose levels. Statistical significance was set at *p* < 0.05.

## 4. Results

### 4.1. Anthropometry

The characteristics of the study participants are shown in Table 1. Twenty-two participants stratified by gender (men: *n* = 11; women: *n* = 11) were included in the study.

### 4.2. Peak Postprandial Glucose Levels Are More Variable than Fasting Glucose Levels

Changes in glucose levels during the 24-h period and fasting period (1 point before breakfast) and peak (after each meal) glucose levels are shown in Figure 1. The results in Figure 1b indicate that the peak glucose levels after each meal had significantly more variance than the glucose levels after fasting (breakfast: *p* < 0.001; lunch: *p* < 0.001; dinner: *p* < 0.006).

### 4.3. Gut Bacteria Data

In this study, only the gut bacteria found in more than half of the participants were used for analysis. The gut bacteria used in the analysis and their abundance in each participant are presented in Table 2. The relative abundance of gut bacteria varied among participants; color intensity increased with increasing abundance. Table 2 depicts the relative abundances of gut bacteria at the phylum and genus levels in each participant. Notably, Firmicutes were highly abundant.

### 4.4. Peak Glucose Levels after Dinner Are Highly Correlated with the Gut Bacteria Compared to Other Postprandial Peak Glucose Levels

The correlation between gut bacteria and the fasting glucose levels and the peak after each meal is shown in Table 3. The most common statistically significant correlation was between the gut bacteria and the peak glucose levels after dinner (20.0%). Six gut bacteria, *Bacteroides*, the *Clostridiales Clostridiaceae* group, *Anaerostipes*, the *Clostridiales [Mogibacteriaceae]* group, *Holdemania*, and *Bilophila* showed a significant correlation only with the peak glucose levels after dinner (Table 3). In contrast, the *Bacteroidales s24-7* group showed a negative correlation with fasting glucose levels and the peak glucose levels after each meal (Table 3).

### 4.5. 4 h AUC after Dinner Are Highly Correlated with the Gut Bacteria Compared to Other Postprandial Peak Glucose Levels

The correlation between gut bacteria and the 4-h AUC after each meal is shown in Table 4. There was a most common statistically significant correlation between the gut bacteria and the 4-h AUC after dinner (14.0%). Six gut bacteria, the *Bacteroidales Rikenellaceae* group, the *Bacteroidales s24-7* group, the *Bacteroidales [Barnesiellaceae]* group, *Blautia*, *Ruminococcus*, the *Clostridiales [Mogibacteriaceae]* group, and *Holdemania* showed a significant correlation with both peak glucose levels after dinner and 4-h AUC after dinner (Table 3 and Table 4).

## 5. Discussion

As a result of the present study, we observed many of the most significant correlations between the gut bacteria and the peak glucose levels after dinner and the 4-h AUC after dinner. In addition, since significant correlations were confirmed mainly at the genus level, it is necessary to look at the level above the genus in order to analyze the gut microbiota.

The six gut microbiota, *Bacteroides*, the *Clostridiales Clostridiaceae* group, the *Anaerostipes*, *Clostridiales [Mogibacteriaceae]* group, *Holdemania*, and *Bilophila* showed a significant correlation only with the peak glucose levels after dinner. Therefore, these gut bacteria can be used as markers for predicting peak glucose levels after dinner. Since undetermined gut bacteria, such as the *Clostridiales Clostridiaceae* group and the *Clostridiales [Mogibacteriaceae]* group are also related, it may be possible to predict glucose levels by focusing on undetermined gut bacteria. On the contrary, the *Bacteroidales s24-7* group showed a negative correlation with fasting glucose levels and the peak glucose levels after each meal. In mice, the *Bacteroidales s24-7* group is associated with bacteria that produce short-chain fatty acids (SCFAs) [18]. SCFAs produced in the intestine promote the secretion of hormones, such as glucagon-like peptide-1 (GLP-1) and peptide YY (PYY), by binding to G-protein-coupled receptors (GPR41 and GPR43), which are SCFA receptors present in colon L cells [19]. GLP-1 binds to GLP-1 receptors present on pancreatic β-cells, promotes insulin secretion, and suppresses elevated blood glucose levels [20]. PYY acts on Y2 receptors in the hypothalamus of the brain and suppresses appetite [21]. Based on the above-mentioned studies and our present results, an increase in the abundance of the *Bacteroidales s24-7* group reduced fasting and postprandial blood glucose levels through the production of SCFAs. However, there are other bacteria that produce SCFAs. For example, Bifidobacterium stimulates the production of acetic acid and butyric acid [22]. Nevertheless, it is unclear why only the *Bacteroidales s24-7* group specifically showed a negative correlation with fasting and postprandial glucose levels in the present study. In future experiments, when we have a chance to measure other blood hormones such as GLP-1 and PYY, it may be possible to further understand the relationship between gut bacteria and glucose levels.

In addition, previous studies have reported that *Akkermansia* is inversely proportional to fasting blood glucose levels [23]. However, no correlation was found between *Akkermansia* and fasting and postprandial blood glucose levels in this study. On the other hand, the amount of *Akkermansia* has been reported to decrease in the elderly [24]. Therefore, it is possible that no correlation was found between *Akkermansia* and fasting and postprandial blood glucose levels in present aging persons. In addition, it may be related to the fact that this study target is Japanese people.

Similarly, in the correlation between AUC and gut bacteria, the most common statistically significant correlation was with the 4-h AUC after dinner. In particular, the six bacteria, the *Bacteroidales Rikenellaceae* group, the *Bacteroidales s24-7* group, the *Bacteroidales [Barnesiellaceae]* group, *Blautia, Ruminococcus*, the *Clostridiales [Mogibacteriaceae]* group, and *Holdemania* were significantly correlated with both peak glucose levels after dinner and 4-h AUC after dinner. Therefore, it is possible that these bacteria are more closely associated with postprandial blood glucose than other gut microbiotas.

A previous study reported that the association of diet, gut microbiota, and blood markers is generally stronger with lipid indicators than with blood glucose indicators [25]. However, this previous study investigated the relationship between blood glucose levels after breakfast or lunch and the gut microbiota, but not the relationship between blood glucose levels after dinner and the gut microbiota. Based on our study, it is necessary to measure the glucose levels after dinner while examining the relationship between glucose levels and the gut microbiota. At present, it is not clear why a correlation with more bacterial species was observed after dinner, but it is probably related to the circadian system and the timing of the three meals. Circadian rhythms control the timings of digestion, absorption, and metabolism in the stomach and intestines; additionally, these circadian systems control glucose tolerance to meals [26,27]. In fact, glucose tolerance is higher in the morning and lower in the evening [28]. Circadian rhythms also influence the composition of the gut microbiota and are controlled by dietary timings. Indeed, the composition of the gut microbiota of mice that were restricted-fed during the active phase and those that were restricted-fed during the inactive phase showed opposite rhythms [29]. In addition, the duration of fasting is important for changes in the composition of the gut microbiota [30]. As there are circadian rhythms for changes in the blood glucose levels and changes in the composition of the gut microbiota, it is believed that these factors interact with each other; as a result, a correlation was found between various gut bacterial species and glucose levels, especially after dinner.

The results of this study showed that there was a high correlation between dinner postprandial glucose levels and various gut bacterial species in comparison with the correlation between fasting glucose levels and the gut microbiota. This indicates that glucose fluctuations after dinner may be predicted by examining the individual pattern of gut microbiota. Moreover, individual microbiota may be able to predict the risk of aberrant glucose metabolism, such as diabetes, because the risk of diabetes is well correlated with the postprandial glucose level, especially after dinner, rather than fasting glucose levels.

## 6. Limitations

Despite these findings, our study had several limitations. First, since the study only focused on older people, the results may not be applicable to young people. Hence, the findings of this study may not be generalizable. Future studies should expand the scope of this study. Second, this study was conducted in Japan, and it is unclear whether the same results would be obtained in other countries. Therefore, the number of target countries needs to be expanded in future studies. Since the composition of microbiota differs greatly among people, it would be beneficial to confirm these findings in a considerably larger population to obtain more accurate results.

## 7. Conclusions

*Bacteroides*, the *Bacteroidales Rikenellaceae* group, the *Bacteroidales s24-7* group, the *Bacteroidales [Barnesiellaceae]* group, *Blautia, Ruminococcus*, the *Clostridiales Clostridiaceae* group, *Anaerostipes*, the *Clostridiales [Mogibacteriaceae]* group, *Holdemania*, and *Bilophila* were correlated with peak blood glucose levels and/or 4-h AUC after dinner and focusing on the composition of these gut microbiota may help to predict not only post-dinner hyperglycemia but disease risk, such as diabetes.

## Figures and Tables

**Figure 1 metabolites-12-00669-f001:**
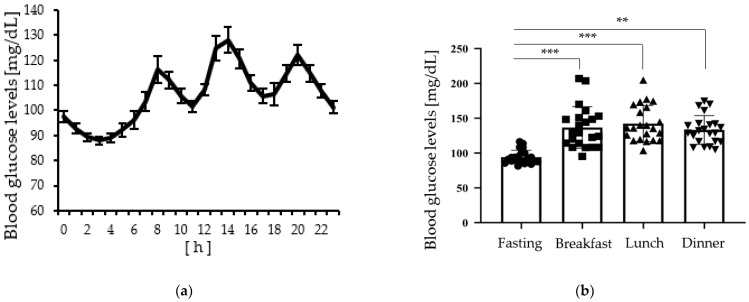
(**a**) Changes in the blood glucose levels for 24 h (*n* = 22). Values are expressed as means and standard errors; (**b**) Comparison of blood glucose levels fasting (1 point before breakfast) and peak after each meal. It shows the average and the blood glucose level of each participant. The coefficients of variation (=CV) were Fasting = 0.101, Breakfast = 0.213, Lunch = 0.177, Dinner = 0.151. ** *p* < 0.01, *** *p* < 0.001, vs. Fasting (Levene’s test).

**Table 1 metabolites-12-00669-t001:** Characteristics of study participants.

Physical Characteristics	All Participants (*n* = 22)	Men (*n* = 11)	Women (*n* = 11)
Age (years)	74.2 ± 5.29	73.7 ± 5.53	74.7 ± 4.99
Height (cm)	159.2 ± 9.64	167.1 ± 5.38	151.3 ± 5.59
Weight (kg)	57.5 ± 10.5	65.2 ± 6.63	49.8 ± 7.51
BMI ^1^ (kg/m^2^)	22.4 ± 2.16	23.3 ± 1.74	21.6 ± 2.21

^1^ BMI: body mass index. All data are presented as the mean ± standard deviation.

**Table 2 metabolites-12-00669-t002:** Composition of the gut microbiota in each participant.

1	2	3	4	5	6	7	8	9	10	11	12	13	14	15	16	17	18	19	20	21	22	
0.009476	0.17847	0.128834	0.208855	0.022017	0.120324	0.029588	0.021804	0.037065	0.025419	0.082006	0.130338	0.190599	0.133793	0.158384	0.158813	0.085217	0.052596	0.283104	0.411319	0.092044	0.130986	*Actinobacteria*
0.372157	0.300095	0.391323	0.367502	0.241431	0.17339	0.286016	0.141513	0.245892	0.318136	0.286785	0.31188	0.242756	0.050115	0.26683	0.152269	0.290848	0.217896	0.112343	0.24002	0.391056	0.216588	*Bacteroidetes*
0.579083	0.473816	0.467309	0.394748	0.603453	0.696259	0.662205	0.820436	0.702713	0.512407	0.630272	0.445052	0.548616	0.796782	0.56989	0.67801	0.604298	0.510246	0.59257	0.317837	0.50494	0.637089	*Firmicutes*
0.029635	0.02537	0.009904	0.018677	0.118589	0.009101	0.016555	0.015177	0.014329	0.13839	0.000937	0.11273	0.012878	0.001839	0.004896	0.006981	0.009263	0.219262	0.005992	0.029813	0.01196	0.015023	*Proteobacteria*
0.000172	0.000475	0	0.00011	0.001001	0.000463	0	0.002779	0.000191	0.000403	0	0	0	0.002299	0	0.001309	0	0	0.000599	0.003032	0.000347	0.000313	*Actinomyces*
0.001034	0.069801	0.080894	0.178532	0.003252	0.117162	0.008101	0.01646	0.000573	0.003631	0.026242	0.096134	0.142949	0.036322	0.149816	0.035777	0.022601	0.022541	0.171959	0.296109	0.08667	0.108138	*Bifidobacterium*
0.000345	0	0.00184	0	0.000751	7.71 × 10^−5^	0.002466	0	0.005923	0	0	0	0.003863	0.002299	0.000979	0.00829	0.007781	0.002732	0.025764	0.001516	0.000173	0.001565	*f__Coriobacteriaceae;g__*
0	0.10426	0.039439	0.020105	0.015512	0	0.018669	0	0.028468	0.020173	0.055295	0.033596	0.042498	0.087816	0	0.113002	0.054094	0.02459	0.080288	0.103588	0	0.020188	*Collinsella*
0.007064	0.003799	0	0.009888	0	0.002622	0	0.001496	0.000191	0.000807	0.000469	0.000607	0	0.00092	0.001224	0	0	0	0.001498	0.007074	0.00468	0	*Eggerthella*
0.347174	0.24515	0.033567	0.350472	0.029022	0.170382	0.254315	0.117572	0.191058	0.141416	0.173383	0.268974	0.065679	0.024368	0.134884	0.04363	0.078177	0.012295	0.038945	0.195048	0.339227	0.180751	*Bacteroides*
0.017746	0.009565	0.013848	0.00835	0.015261	0	0.004227	0.008337	0.014329	0.024208	0.025305	0.034406	0.009015	0.004598	0.005875	0.01178	0.016673	0.004098	0.017675	0.044467	0.049575	0.00626	*Parabacteroides*
0	0.033238	0.33567	0.000659	0.178384	0	0	0.000641	0.006496	0.140408	0.079663	0.000405	0.156471	0.001839	0.113097	0.003054	0.178585	0.193306	0.043439	0.000505	0.00052	0	*Prevotella*
0	0.001832	0.000613	0.006482	0.005254	0.002468	0.007397	0.008764	0.002866	0.010087	0.005155	0.007084	0.002576	0.013333	0.001469	0.027051	0.001482	0.002049	0.002097	0	0.000173	0.006416	*f__Rikenellaceae;g__*
0.000345	0.000204	0.000964	0.000439	0.005504	0	0.001761	0.00342	0.000573	0.000202	0.000469	0.000607	0.003863	0.002759	0	0.036213	0	0.000683	0.004793	0	0.00052	0.000156	*f__S24-7;g__*
0	0	8.76 × 10^−5^	0.00011	0.003252	0	0	0	0.00535	0	0	0	0.001288	0.001379	0.003672	0.001309	0.007781	0	0.001498	0	0	0.002817	*f__[Barnesiellaceae];g__*
0.006547	0.004545	0.001315	0.000989	0.002002	0.000463	0.002466	0.001924	0.000382	0.001816	0.000469	0	0.001288	0.001839	0	0.000436	0.001112	0.000683	0.000899	0	0.00052	0.003756	*Odoribacter*
0.005341	0.001764	0.001227	0.000659	0.027521	0.000309	0.000704	0.189825	0.001528	0.002824	0.001406	0.002024	0.001288	0.234023	0.000245	0.039267	0.000371	0.028689	0.0003	0	0.001213	0.000626	*Lactobacillus*
0.000861	0.000611	0.011481	0.00033	0.005754	0.000154	0.001057	0.001496	0.000191	0.000403	0.000937	0.001012	0.001288	0.00046	0.000245	0.000436	0.000371	0	0.000599	0.001011	0.00156	0.000313	*Lactococcus*
0.037388	0.002849	0.000876	0.00022	0.070303	0.038257	0.00317	0.141727	0.012228	0.047811	0.044049	0.024691	0	0.204598	0.132436	0.000436	0.002594	0.109973	0.006591	0.004548	0.001907	0.008607	*Streptococcus*
0.056513	0.017637	0.01823	0.013294	0.031524	0.022059	0.023952	0.037837	0.044899	0.044987	0.052015	0.011536	0.052157	0.037701	0.040881	0.167103	0.057799	0.028005	0.059916	0.009096	0.023921	0.022222	*o__Clostridiales;f__;g__*
0.006892	0	0.000175	0.000879	0.001501	0.000309	0.000352	0.003206	0	0.000403	0.000937	0	0.007083	0.007816	0	0.010908	0.000741	0	0.001797	0	0	0	*f__Clostridiaceae;g__*
0.000172	0	0.000351	0.00022	0.00025	0.000926	0	0.000641	0.001146	0.002824	0.007498	0.001012	0.000644	0	0.002203	0.000436	0.001482	0	0.002097	0	0.002947	0.001095	*Clostridium*
0.012922	0.000339	8.76 × 10^−5^	0	0.00025	0.002005	0.005284	0	0	0	0.000937	0.000202	0.000644	0.00046	0	0.000873	0.002223	0.000683	0.000599	0.000505	0	0	*SMB53*
0.195382	0.094899	0.093339	0.075258	0.078559	0.21558	0.141247	0.098974	0.16603	0.073432	0.156045	0.205829	0.090148	0.022069	0.082742	0.02356	0.093738	0.060792	0.04973	0.023749	0.119258	0.182942	*f__Lachnospiraceae;g__*
0.00224	0.001492	0.001139	0.000989	0.001001	0.002391	0.002113	0.008551	0.009935	0.002421	0.001874	0.005262	0.000644	0	0.026928	0	0.001853	0.003415	0	0.00859	0.002427	0.004851	*Anaerostipes*
0.056513	0.119319	0.046012	0.022962	0.025269	0.049749	0.05037	0.014964	0.055216	0.020779	0.086223	0.014572	0.03284	0.027586	0.0612	0.013089	0.033716	0.036202	0.046135	0.051541	0.07263	0.053678	*Blautia*
0.000517	0.004545	0.005171	0.002966	0.002252	0.02499	0.005988	0.007054	0.015285	0.004438	0.020619	0.003441	0.006439	0.022529	0.010282	0.031414	0.050389	0.010246	0.031156	0	0.050789	0.020031	*Coprococcus*
0.007753	0.027676	0.01078	0.012854	0.009757	0.011338	0.016203	0.007268	0.023118	0.002219	0.013121	0.002429	0.005151	0.008736	0.010037	0.003927	0.008892	0.009563	0.009587	0.039414	0.001213	0.011737	*Dorea*
0.029118	0.009768	0.007099	0.001868	0.017013	0.025916	0.002818	0.023728	0.022927	0.022594	0.007966	0.009917	0.013522	0.003218	0.000979	0.004799	0.039644	0.010929	0.010485	0.006064	0.015947	0.019249	*Lachnospira*
0.002757	0.002442	0.010079	0.000989	0.006505	0.002854	0.00317	0.031424	0.002675	0.031874	0.005623	0.031573	0.001932	0	0.001224	0.001309	0.01482	0.006148	0.012582	0.006569	0.00104	0.012989	*Roseburia*
0.009649	0.022996	0.002366	0.054274	0.003252	0.011955	0.005636	0.002565	0.017959	0.007666	0.010309	0.007893	0.001288	0.008736	0.003672	0	0.00741	0.000683	0.004494	0.028802	0.007454	0.001565	*[Ruminococcus]*
0	0.000136	0	0.00011	0.001001	0.000154	0	0.003848	0.000764	0.002623	0	0.000202	0.000644	0.00092	0	0	0	0	0	0.001516	0	0	*[Clostridium]*
0.077016	0.02442	0.04645	0.041529	0.018764	0.070729	0.130328	0.061992	0.049102	0.075852	0.082474	0.051407	0.061816	0.078621	0.082252	0.114311	0.08781	0.036885	0.131516	0.009096	0.074536	0.075743	*f__Ruminococcaceae;g__*
0.026878	0.035206	0.050394	0.006482	0.044533	0.148708	0.108489	0.089141	0.05961	0.048416	0.082943	0.038251	0.101095	0.010115	0.015177	0.024433	0.063727	0.065574	0.020971	0.033855	0.082163	0.091236	*Faecalibacterium*
0.023604	0.016484	0.005171	0.035597	0.003252	0.004165	0.027122	0.00513	0.018151	0.009482	0.008435	0.005869	0.008371	0.006897	0.013219	0.012216	0.00741	0.002049	0.009287	0.015664	0.015774	0.009077	*Oscillospira*
0.001895	0.0097	0.00929	0.065041	0.047035	0.021751	0.094752	0.04938	0.097822	0.052249	0.009841	0.001214	0.057952	0.068966	0.013709	0.106457	0.05817	0.003415	0.088975	0.002021	0.010574	0.068858	*Ruminococcus*
0.01275	0.005766	8.76E-05	0.013733	0	0.004782	0	0.001283	0	0.041356	0	0.016191	0	0.001839	0	0.016579	0.029641	0.004098	0	0	0	0	*Dialister*
0	0.022317	0.02603	0.002857	0.016763	0.011492	0.013385	0.000214	0.009553	0	0.030928	0.00081	0.027688	0.001839	0.035985	0.003054	0.011115	0.015027	0.026363	0.006569	0	0.012989	*Phascolarctobacterium*
0	6.78 × 10^−5^	0	0	0.006255	0.002005	0	0.024583	0	0.005649	0	0	0	0.001379	0	0	0.010004	0.000683	0	0.004042	0.001387	0.001095	*Veillonella*
0	0.001153	0.001139	0.000769	0.001001	7.71 × 10^−5^	0.001761	0	0.002675	0.000807	0.000937	0.000607	0.001288	0.001839	0.001224	0.006981	0.001482	0.000683	0.006291	0	0.000173	0.001095	*f__[Mogibacteriaceae];g__*
0.004307	0.04572	0.000438	0.001318	0.003002	0.004936	0.001409	0.001496	0.000382	0.008069	0.002343	0.000202	0.01159	0	0.018605	0.006108	0.009633	0.002732	0.055722	0.009601	0.014734	0.032238	*f__Erysipelotrichaceae;g__*
0.000172	0.000271	0.000526	0.00011	0.00025	0.000231	0	0.002779	0.000573	0	0	0.000607	0	0.001379	0.000245	0.000873	0	0	0	0	0.000693	0	*Allobaculum*
0.002068	0.001425	0.001052	0.000549	0.00025	0.000848	0.001761	0.000214	0.000191	0	0.000469	0	0.000644	0	0.014688	0.000436	0.000371	0	0.0003	0.015664	0	0.000313	*Coprobacillus*
0	0	0.000438	0.000769	0.00025	0.000694	0.001761	0	0.000382	0.001009	0	0.000202	0	0	0	0.000436	0	0	0.000599	0	0	0.000469	*Holdemania*
0.004997	0.001153	0.000526	0.001868	0.042782	0.001697	0.000704	0	0.000955	0.000202	0.000469	0.000405	0.041211	0	0.001224	0.002182	0	0.043033	0.000599	0.011622	0.001907	0.000156	*[Eubacterium]*
0	0.019604	0.002542	0.012085	0.003252	0.005245	0.001761	0.00513	0.010508	0.003833	0	0.011131	0.005151	0.001379	0	0.000436	0.004817	0.008197	0.000899	0	0.006414	0.011268	*Sutterella*
0.005169	0.004274	0.00149	0.001538	0.00025	0.00108	0.000352	0.000428	0.003248	0	0	0.002631	0	0	0	0.000873	0.000741	0	0.001797	0.006064	0.003813	0.001252	*Bilophila*
0.001551	0.001153	0.004996	0.001538	0.063548	0.001774	0.00634	0.001924	0.000191	0.002017	0.000937	0.078122	0.003863	0	0.004896	0.001745	0.001482	0.209016	0.001797	0.011622	0	0.001408	*f__Enterobacteriaceae;g__*
0.016196	6.78 × 10^−5^	0.000701	0	0.014261	0.000231	0.004579	0.000855	0	0.12326	0	0	0.001932	0	0	0.000873	0.001853	0	0	0	0	0.000626	*Erwinia*
0.009132	0.003799	0.002629	0.00791	0.014511	0.000771	0.004931	0.000641	0	0.004842	0	0	0.004507	0.016092	0	0	0.009633	0	0.005392	0	0	0.000156	*Akkermansia*

Only the gut bacteria used in the analysis in this study are shown. The higher the abundance, the darker the red color.

**Table 3 metabolites-12-00669-t003:** Correlation between gut bacteria and blood glucose levels of fasting and peak after each meal.

Fasting	Breakfast	Lunch	Dinner	
				*Actinobacteria*
	#			*Bacteroidetes*
				*Firmicutes*
				*Proteobacteria*
				*Actinomyces*
				*Bifidobacterium*
				*f__Coriobacteriaceae;g__*
				*Collinsella*
				*Eggerthella*
			#	*Bacteroides*
				*Parabacteroides*
				*Prevotella*
	**		**	*f__Rikenellaceae;g__*
*	*	*	**	*f__S24-7;g__*
				*f__[Barnesiellaceae];g__*
				*Odoribacter*
				*Lactobacillus*
				*Lactococcus*
				*Streptococcus*
				*o__Clostridiales;f__;g__*
			*	*f__Clostridiaceae;g__*
				*Clostridium*
				*SMB53*
				*f__Lachnospiraceae;g__*
			*	*Anaerostipes*
#	#		#	*Blautia*
				*Coprococcus*
				*Dorea*
				*Lachnospira*
				*Roseburia*
				*[Ruminococcus]*
		**		*[Clostridium]*
				*f__Ruminococcaceae;g__*
				*Faecalibacterium*
				*Oscillospira*
	*		**	*Ruminococcus*
				*Dialister*
				*Phascolarctobacterium*
				*Veillonella*
			**	*f__[Mogibacteriaceae];g__*
				*f__Erysipelotrichaceae;g__*
				*Allobaculum*
				*Coprobacillus*
			*	*Holdemania*
				*[Eubacterium]*
				*Sutterella*
			*	*Bilophila*
				*f__Enterobacteriaceae;g__*
				*Erwinia*
				*Akkermansia*

Positive correlations are shown in red, and negative correlations are shown in green. * *p* < 0.05, ** *p* < 0.01 (Spearman correlation), # *p* < 0.05 (Pearson correlation).

**Table 4 metabolites-12-00669-t004:** Correlation between gut bacteria and blood glucose levels of 4-hour AUC after each meal.

Breakfast	Lunch	Dinner	
			*Actinobacteria*
			*Bacteroidetes*
			*Firmicutes*
			*Proteobacteria*
			*Actinomyces*
			*Bifidobacterium*
			*f__Coriobacteriaceae;g__*
			*Collinsella*
			*Eggerthella*
			*Bacteroides*
			*Parabacteroides*
			*Prevotella*
	*	*	*f__Rikenellaceae;g__*
		**	*f__S24-7;g__*
		*	*f__[Barnesiellaceae];g__*
			*Odoribacter*
			*Lactobacillus*
			*Lactococcus*
			*Streptococcus*
			*o__Clostridiales;f__;g__*
			*f__Clostridiaceae;g__*
			*Clostridium*
			*SMB53*
			*f__Lachnospiraceae;g__*
			*Anaerostipes*
		#	*Blautia*
			*Coprococcus*
			*Dorea*
			*Lachnospira*
			*Roseburia*
			*[Ruminococcus]*
	**		*[Clostridium]*
			*f__Ruminococcaceae;g__*
			*Faecalibacterium*
			*Oscillospira*
*		**	*Ruminococcus*
			*Dialister*
			*Phascolarctobacterium*
			*Veillonella*
		*	*f__[Mogibacteriaceae];g__*
			*f__Erysipelotrichaceae;g__*
			*Allobaculum*
			*Coprobacillus*
		*	*Holdemania*
			*[Eubacterium]*
			*Sutterella*
			*Bilophila*
			*f__Enterobacteriaceae;g__*
			*Erwinia*
			*Akkermansia*

Positive correlations are shown in red, and negative correlations are shown in green. * *p* < 0.05, ** *p* < 0.01 (Spearman correlation), # *p* < 0.05 (Pearson correlation).

## Data Availability

Data will be sent on request from the corresponding author. The data are not publicly available due to privacy restrictions.

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
