# Peer review of "Relationship between Fasting and Postprandial Glucose Levels and the Gut Microbiota"

_metabolites, 2022, doi:10.3390/metabo12070669_

Round 1
Reviewer 1 Report
The manuscript by Yui Mineshita et al. investigated the relationship between fasting and postprandial glucose levels and the gut microbiota. The study is of interest to readers. I have the following suggstions and comments:
1, Figure 2 and figure 3 should be in a table format, not a figure format. Additionally, The authors must specify whether it is Pearson correlation or Spearman correlation? The authors could not see it is Pearson correlation or Spearman correlation. I would confuse the readers.
2, Akkermansia has been demonstrated to be correlated with changes of the serum glucose levels. However, the findings of the present study do not support such a claim. Why? This should be discussed.
3, The conlusion part should be revised. For example, which bacterium are correlated with changes of the glucose levels?
4, the English of the manuscript must be further improved.
Author Response
I really appreciate all the changes that have been done in accordance with my comments and suggestion. However, I still found that the manuscript needs to be improved and better detailed in some parts. Please see my comments below:
Query 1:
Figure 2 and figure 3 should be in a table format, not a figure format. Additionally, The authors must specify whether it is Pearson correlation or Spearman correlation? The authors could not see it is Pearson correlation or Spearman correlation. I would confuse the readers.
Response 1:
Thank you for your comments.
As you suggested, we changed the figure format (figure 2 and figure 3) to a table format (table 3 and table 4). We also used different notations for Pearson and Spearman in table 3 and table 4.
Query 2:
Akkermansia has been demonstrated to be correlated with changes of the serum glucose levels. However, the findings of the present study do not support such a claim. Why? This should be discussed.
Response 2:
Thank you for your comments.
As you suggested, there is an association between Akkermansia and blood glucose levels. However, we believe that no correlation was confirmed in the results of this study because the subjects of this study are elderly people, and the races of the subjects are different.
We added to the Discussion (line 226-232).
Query 3:
The conclusion part should be revised. For example, which bacterium are correlated with changes of the glucose levels?
Response 3:
Thank you for your comments.
As you suggested, in the Conclusion part, we mentioned the correlation between bacteria and blood glucose levels and added that in the future, observing these bacteria may help predict the risk of developing type 2 diabetes and other diseases.
Query 4:
the English of the manuscript must be further improved.
Response 4:
Thank you for your comments.
As you suggested, we improved the English composition of the manuscript. Please check it.
Reviewer 2 Report
Comments of the manuscript: Manuscript ID: metabolites-1807637
Thank you for the opportunity to review the article. Authors are advised to address the following comments to improve the quality of the manuscript in its current form:
1. At the methodology, the use of tissue glucose level in ‘Determination of tissue glucose levels’ is confusing! The instrument still measures blood glucose level.
2. Could the results of this study be translated to the diabetic patients? If yes how?
3. Don’t you think what the subjects ate may influence the data of this study? This is why fasting is the best than postprandial…
4. Why insulin is not included as a factor?
Thank you.
Author Response
I really appreciate all the changes that have been done in accordance with my comments and suggestion. However, I still found that the manuscript needs to be improved and better detailed in some parts. Please see my comments below:
Query 1: At the methodology, the use of tissue glucose level in ‘Determination of tissue glucose levels’ is confusing! The instrument still measures blood glucose level.
Response 1:
Thank you for your precise remarks. The FreeStyle Libre Pro used in this study measures the concentration of glucose in the interstitial fluid. Interstitial fluid is the fluid that exists between cells. Glucose is transported throughout the body through the blood vessels, from the capillaries to the interstitial fluid, and from the interstitial fluid to the cells for use. The glucose concentration in the blood flowing through the blood vessels is the "blood glucose level," and it has been proven that there is a high correlation between the blood glucose level and the glucose level in the interstitial fluid (Rebrin K, Steil GM. Diabetes Technol Ther. 2000; 2(3): 461-472.). Because of this, we referred to tissue glucose in the text, but as you pointed out, it accurately refers to the concentration of glucose in the interstitial fluid, so we have corrected the notation in the text.
Query 2: Could the results of this study be translated to the diabetic patients? If yes how?
Response 2:
This study was conducted on healthy elderly people participate. It is also known that the intestinal microbiota of diabetic patients differs from that of healthy individuals. Therefore, it cannot be determined whether the results of this study can be translated to diabetic patients. In the future, as you suggest, we will conduct more research on diabetic patients and other patients.
Query 3: Don’t you think what the subjects ate may influence the data of this study? This is why fasting is the best than postprandial…
Response 3:
As you suggested, changes in gut bacteria are related to many factors, and we believe that it is necessary to consider the relationship with other factors in the future.
However, since the most significant correlations were between the gut bacteria and the peak glucose levels after dinner and the 4-hour AUC after dinner rather than fasting blood glucose levels, there was some relationship between the blood glucose levels after dinner and the gut bacteria. Therefore, we considered it to be one of the predictors of the gut microbiota.
In addition, since the postprandial blood glucose level varied more than the fasting blood glucose level, it is considered that the postprandial blood glucose level may be a better predictor of the gut microbiota.
Query 4: Why insulin is not included as a factor?
Response 4:
Thank you very much for your suggestions. There is no insulin data because the protocol for this study does not include blood sampling. As you pointed out, insulin sensitivity is closely related to microbiota (Kimura I et al., Nat Commun. 2013:4:1829). In addition, based on your point, we will consider insulin monitoring as an issue for future study.
Reviewer 3 Report
The paper evaluated the relationship between fasting and postprandial glucose levels and the gut microbiota. Some questions are as follows:
1. Line16:lev-els
2. The title of Table 2 and Table 3 was same (also 4.3 and 4.2 ), authors could consider synthesizing a single table or indicating whether each table refers to different categories.
3. Why the authors analyzed the relationship between the microbiota and different types of blood glucose levels? Abstract and Introduction need to be rewritten, the expression is not clear and the logic is confused..
4. The conclusions are unreliable. (1) Although hypertension is common in the elderly population, but why choose elderly people with hypertension as the subject of study? The results are only representative of elderly patients with hypertension, and both the conclusions and the title need to be revised. (2) Changes in gut bacteria are related to many factors, and the author simply analyzed the correlation between its abundance and blood glucose. Although the significant correlations between the gut bacteria and the peak glucose levels after dinner and the 4-hour AUC after dinner were observed, the conclusion of “the glucose levels after dinner are a better predictor of microbiota conditions than fasting glucose levels” was inappropriate.
5. Line 55: “aged 65” Average Age? Values should expressed as Mean ±SD
6. Table 1 :“All data are presented as the mean ± standard error” should be statistical analysis as Standard Deviation. Meanwhile, table 1 was not statistically analyzed. Are there statistical differences in age among the patients selected for the study?
Author Response
I really appreciate all the changes that have been done in accordance with my comments and suggestion. However, I still found that the manuscript needs to be improved and better detailed in some parts. Please see my comments below:
Query 1:
Line66: lev-els
Response 1:
Thank you for your comments. We have rewritten this sentence.
Query 2:
The title of Table 2 and Table 3 was same (also 4.3 and 4.2 ), authors could consider synthesizing a single table or indicating whether each table refers to different categories.
Response 2:
Thank you for your comments.
As you suggested, we synthesized a single table (table 2) and changed title of 4.3.
Query 3:
Why the authors analyzed the relationship between the microbiota and different types of blood glucose levels? Abstract and Introduction need to be rewritten, the expression is not clear and the logic is confused.
Response 3:
Thank you for your comments.
As you suggested, I have also rewritten the Abstract and Introduction and Conclusion to clearly state that in the future, observing the gut microbiota will be used to help to predict the risk of diseases such as type 2 diabetes (lines 22-23, 48-53 and 277-282).
Query 4:
The conclusions are unreliable. (1) Although hypertension is common in the elderly population, but why choose elderly people with hypertension as the subject of study? The results are only representative of elderly patients with hypertension, and both the conclusions and the title need to be revised. (2) Changes in gut bacteria are related to many factors, and the author simply analyzed the correlation between its abundance and blood glucose. Although the significant correlations between the gut bacteria and the peak glucose levels after dinner and the 4-hour AUC after dinner were observed, the conclusion of “the glucose levels after dinner are a better predictor of microbiota conditions than fasting glucose levels” was inappropriate.
Response 4:
Thank you for your comments.
(1) : In this study, elderly people with hypertension are excluded from the study. Therefore, only healthy elderly people participate in this study. I'm sorry it's hard to understand. We changed the text (line 60).
(2) : As you suggested, changes in gut bacteria are related to many factors, and we believe that it is necessary to consider the relationship with other factors in the future.
However, since the most significant correlations were between the gut bacteria and the peak glucose levels after dinner and the 4-hour AUC after dinner, there was some relationship between the blood glucose levels after dinner and the gut bacteria. Therefore, we considered it to be one of the predictors of the gut microbiota.
In addition, since the postprandial blood glucose level varied more than the fasting blood glucose level, it is considered that the postprandial blood glucose level may be a better predictor of the gut microbiota.
Query 5:
Line 55: “aged 65” Average Age? Values should expressed as Mean ±SD
Response 5:
Thank you for your comments. It is not the average. It means over 65 years old.
We have rewritten this sentence (lines 57).
Query 6:
Table 1 :“All data are presented as the mean ± standard error” should be statistical analysis as Standard Deviation. Meanwhile, table 1 was not statistically analyzed. Are there statistical differences in age among the patients selected for the study?
Response 6:
Thank you for your comments.
In this study, we could not perform statistical analysis because we did not compare by grouping. However, since it is intended for healthy elderly people aged 65 and over, we believe that the difference in age is not large (<standard value ± 2SD).
As you suggested, we changed the standard error to the standard deviation (table 1, lines 145).
Round 2
Reviewer 1 Report
The authors have revised the manuscript accordingly. I sugges to accept this manuscript.
Reviewer 3 Report
No comments.